

# Different responses of plant N and P resorption to overgrazing in three dominant species in a typical steppe of Inner Mongolia, China

Zhen Wang[1], Saheed Olaide Jimoh[1,2], Xiliang Li[1], Baoming Ji[3], Paul C. Struik[4], Shixian Sun[1], Ji Lei[1], Yong Ding[1] and Yong Zhang[1]

[1] Institute of Grassland Research, Chinese Academy of Agricultural Sciences, Hohhot, Inner Mongolia, China
[2] Sustainable Environment Food and Agriculture Initiative (SEFAAI), Lagos, Nigeria
[3] College of Forestry, Beijing Forestry University, Beijing, China
[4] Department of Plant Sciences, Wageningen University, Wageningen, Netherlands

## ABSTRACT

Nutrient resorption from senesced leaves is an important mechanism for nutrient conservation in plants. However, little is known about the effect of grazing on plant nutrient resorption from senesced leaves, especially in semiarid ecosystems. Here, we evaluated the effects of grazing on N and P resorption in the three most dominant grass species in a typical steppe in northern China. We identified the key pathways of grazing-induced effects on N and P resorption efficiency. Grazing increased N and P concentrations in the green leaves of *Leymus chinensis* and *Stipa grandis* but not in *Cleistogenes squarrosa*. Both *L. chinensis* and *S. grandis* exhibited an increasing trend of leaf N resorption, whereas *C. squarrosa* recorded a decline in both leaf N and P resorption efficiency under grazing. Structural equation models showed that grazing is the primary driver of the changes in N resorption efficiency of the three dominant grass species. For *L. chinensis*, the P concentration in green and senesced leaves increased the P resorption efficiency, whereas the senesced leaf P concentration played an important role in the P resorption efficiency of *C. squarrosa*. Grazing directly drove the change in P resorption efficiency of *S. grandis*. Our results suggest that large variations in nutrient resorption patterns among plant species depend on leaf nutritional status and nutrient-use strategies under overgrazing, and indicate that overgrazing may have indirect effects on plant-mediated nutrient cycling via inducing shifts in the dominance of the three plant species.

# INTRODUCTION

Plant growth is limited by nutrient availability and uptake in various ecosystems (*Chapin III, 1980*). Leaf nutrient resorption is a key process that controls nutrient fluxes from plants to soil and the nutrients available for storage and reuse (*Millard & Proe, 1993*). When plants reuse these nutrients, it improves their nutrient retention and adaptability to the

Corresponding authors
Yong Ding, dingyong228@126.com
Yong Zhang, zycaas@126.com

environment, and reduces their dependence on the nutrient supply from the current environment (*Aerts & Chapin, 2000*; *Lü et al., 2012*). An estimated 50% of total leaf N is resorbed into plant seeds during leaf senescence (*Yuan & Chen, 2009*). The process of nutrient resorption exerts a strong influence on carbon and nutrient cycling (*Kozovits et al., 2007*). Many studies have reported the effect of climate change factors such as temperature (*Yuan & Chen, 2009*), nitrogen addition (*Lü et al., 2013*) and precipitation (*Lü et al., 2012*) on nutrient resorption in plants. Research has also demonstrated that grazing, one of the prominent forms of land use in the pastoral areas, affects plant growth (*Li et al., 2015*) and further impacts soil resource availability and plant nutrient status (*Wang et al., 2014*). Changes in the biogeochemical cycles of grassland ecosystems may have significant implications for plant nutrient resorption (*Lü et al., 2013*; *Ngatia et al., 2015*). However, there is a paucity of information on how grazing impacts nutrient resorption in plants and this study sought to fill this knowledge gap.

Moreover, the effect of grazing on plant diversity, community composition, and ecosystem functioning is well documented in the literature (*Koerner & Collins, 2014*; *Liu et al., 2015*). Over the past 50 years, varying levels of degradation have been reported on Chinese grasslands due to overgrazing (*Wen et al., 2013*). Overgrazing inhibits the growth of palatable and nutrient-rich species with high nutritive value and promotes the dominance of nutrient-poor species or inedible pastures. This phenomenon leads to a decline in nutrient cycling and consequently results in grassland degradation (*Ritchie, Tilman & Knops, 1998*; *Bai et al., 2012*). Overgrazing potentially alters plant nutrient resorption through different mechanisms (*Lü et al., 2015*; *Millett & Edmondson, 2015*; *Ngatia et al., 2015*). Firstly, leaf nutrient resorption is impacted by livestock grazing (*Millett et al., 2005*) depending on the plant's growth stage (*Yasumura et al., 2005*), thereby changing the plant community structure and productivity, soil microbial communities, as well as the physical and chemical properties of the soil (*Li et al., 2008*; *He et al., 2011*). However, the outcome and magnitude of the grazing effect on plant nutrient resorption processes remain unclear (*Ngatia et al., 2015*). Secondly, the preference of livestock for different forage species, trampling of the soil, and the return of excreta to the soil alters plant N and P concentrations (*Heyburn et al., 2017*), which invariably influence plant nutrient resorption. Thirdly, grazing influence nutrient resorption by altering plant phenology, leaf chemistry, and the timing of litterfall which determines the amount of organic matter that is returned back into the soil (*Chapman et al., 2006*). Moreover, long-term grazing reduces the accumulation of litter, which leads to a reduction in soil water content (*Hou et al., 2019*). The change in soil water content has an important effect on plant N and P concentrations (*Bai et al., 2012*) which regulate plant nutrient resorption (*Minoletti & Boerner, 1994*). Notably, soil water availability has a positive effect on soil N transformation and availability (*Wang et al., 2006*).

The responses of plant nutrient resorption to overgrazing is dependent on many complex and connected processes, which makes it difficult to predict leaf N and P resorption in the context of land use and management. *L. chinensis* (tall, perennial C$_3$ rhizome grass), *S. grandis* (tall, perennial C$_3$ bunchgrass), and *C. squarrosa* (short, perennial C$_4$ bunchgrass) are three dominant species distributed widely in the typical steppe of the Mongolian plateau.

The effects of overgrazing on different species result from complex interactions among forage production, quality and phenology (*Ehleringer & Monson, 1993*). Previous studies have shown that overgrazing strongly inhibits the growth of *L. chinensis* and *S. grandis* (*Liang et al., 2009*; *Xie & Wittig, 2003*), whereas *C. squarrosa* is relatively tolerant to grazing (*Liang, Michalk & Millar, 2002*). In this study, we investigate how long-term overgrazing affects the plants' leaf nutrient status and nutrient resorption in three dominant species in the semiarid grasslands of northern China. Specifically, the objectives of this study are to understand: (1) the effects of long-term overgrazing on N and P resorption efficiency and their relationships with plant nutrient concentrations; and (2) the patterns of N and P resorption in three plant species with contrasting overgrazing regulation strategies.

## MATERIALS AND METHODS

### Study sites and field sampling

The study area was conducted at the Inner Mongolia Grassland Ecosystem Research Station (43°38′N, 116°42′E), located in a typical steppe on the Baiyinxile Ranch, Xilinguole in Inner Mongolia, China. The mean annual temperature is 0.7 °C, and the mean annual precipitation is approximately 350 mm, which occurs mainly in the summer period from June to August. The annual precipitation in 2013 (273.4 mm) and 2014 (256 mm) were lower than the long-term average (278.7 mm). The growing season runs from early April to late September for perennial plant species. The dominant species in the plant community are *Leymus chinensis* (Trin.) Tzvel., *Stipa grandis* P. Smirn., and *Cleistogenes squarrosa* (Trin. ex Ledeb.) Keng, which account for 60–80% of the total biomass in the sward. The sub-dominant species are Keng, *Achnatherum sibiricum* (Linn.) Keng, *Agropyron cristatum* (L.) P. Beauv., *Caragana microphylla* Lam., *Artemisia commutata* Besser, *Carex korshinskyi* Kom., *Kochia prostrata* (L.) Schrad., *Serratula centauroides* L., *Koeleria cristata* (L.) Pers., *Artemisia frigida* Willd. Sp. Pl. and *Potentilla bifurca* L. This community is representative of one of the most widely distributed grasslands in the Eurasian steppes. The soils are classified as calcic chernozems (IUSS Working Group WRB, 2006).

### Experimental design

We used long-term freely grazed land and a fenced-off land (i.e., grazing exclusion plot) that were established in 1983. At the time of grazing exclusion, the site was considered to be in excellent condition and representative of an undisturbed community. The grazing plot (∼200 ha in area) is located adjacent to the grazing exclusion plot. The plot has been subjected to grazing by ∼600 sheep and goats year-round for more than 30 years at a stocking rate of ∼3 sheep units per hectare. This stocking rate exceeds the local stocking rate of 1.5 sheep units per hectare recommended by the local government for the maintenance of grass-livestock balance.

### Plant sampling and measurement

In this study, we adopted pseudoreplication and space-for-time substitution limitation (*Hurlbert, 1984*; *Walker et al., 2010*; *Blois et al., 2013*; *Lü et al., 2014*). The sampling area for the grazed and exclusion treatments was 20 × 20 m, established in pairs along transects.

There were fifteen replications for each treatment and the plots were allocated 10 m apart along transects. Prior to the commencement of grazing in each grazed plot, three temporary movable exclusion cages ($1.5 \times 1.5$ m) were set up at each sampling point before the growing seasons in early April 2013 and 2014. Three 1-m$^2$ quadrats were established in the three temporary movable exclusion cages for field investigation and sampling. Subsequently, three 1-m$^2$ quadrats were used to collect samples for evaluating nutrient resorption efficiency and measure aboveground net primary productivity (ANPP).

During the peak growing period (middle of August 2013 and 2014), representative mature green leaves (the ten visible leaves from the top of the shoot) of three dominant species (*L. chinensis*, *S. grandis* and *C. squarrosa*) were sampled in three 1-m$^2$ quadrats. We used the criterion described by *Wright & Westoby (2003)* to determine leaves ready to abscise. In the middle of October 2013 and 2014, we collected the same number of senesced leaves (ten recently senesced leaves in three 1-m$^2$ quadrats) following the aforementioned procedure. The sampled leaves had no obvious leaf area losses and were transported to the laboratory for analysis. The samples were oven-dried at 70 °C for 48 h and subsequently weighed to obtain sample dry weight. The leaves of each species were bulked per plot and passed through a 40 mm mesh screen using a mechanical mill.

All living vascular plants were clipped (1-m$^2$ quadrats) to the ground level for the measurement of ANPP in the current year in each treatment. The collected samples were sorted into species, dried at 70 °C for 48 h in the laboratory and weighed for each quadrat separately. ANPP was calculated as the sum of the aboveground biomass of all vascular plant species based on quadrat.

## Soil sampling and physicochemical properties analysis

We collected soil samples to a depth of 20 cm using an auger (7 cm in diameter and 20 cm long) after the litter layer was removed. Soil organic C (SOC) was determined using the dichromate oxidation method (*Nelson & Sommers, 1982*). Soil total nitrogen (STN) was measured using micro-Kjeldahl digestion (*Nelson & Sommers, 1980*). The $NH_4^+$ and $NO_3^-$ concentrations were measured using a continuous-flow auto-analyzer (Alpkem, OI Analytical, USA) after sample extraction by 2 M KCl at a soil: KCl ratio of 1:5 (w:v). Soil available phosphorus (SAP) was determined using the Kelowna method as described by *Van Lierop (1988)* with a solid to liquid ratio of 1:5. The P concentration in the extracted solution was determined using an Astoria auto-analyzer (Clackamas, OR, Oregon, USA).

## Calculations and statistical analysis

According to *Aerts et al. (2007)*, nitrogen resorption efficiency (NRE) and phosphorus resorption efficiency (PRE) can be calculated based on the total N and P pool in green and senesced leaves. Given that we collected the same number of green and senesced leaves for each species in each plot, nutrient resorption efficiency (RE) was calculated as follows:

$$\text{NRE} = [1 - (\text{Nutrient}_{\text{senesced}}/\text{Nutrient}_{\text{green}})] \times 100\%$$

Where $\text{Nutrient}_{\text{senesced}}$ and $\text{Nutrient}_{\text{green}}$ are the N and P pool in the senesced leaves collected in mid-October 2008 and the green leaves sampled in mid-August 2008, respectively. The values obtained for N and P in each species were multiplied by the

respective biomass to obtain the species N and P concentrations per unit area ($1 \times 1$ m). Resorption efficiency is the level to which the nutrient concentrations are reduced in the senesced leaves (*Killingbeck, 1996*; *Ratnam et al., 2008*). We quantified nutrient resorption efficiency based on the nutrient concentration of the senesced leaves (*Killingbeck, 1996*), with a lower leaf nutrient concentration indicating a higher nutrient resorption efficiency

## Data analysis

Three-way ANOVA was used to examine the effects of species, treatment, year, and their possible interactions on the green and senesced leaf N contents and N resorption efficiency. We used two-way ANOVA to examine the effects of year and treatment on the soil water content (SWC), SOC, STN, soil total phosphorus (STP), soil inorganic nitrogen (SIN: the sum of $NH_4^+$ and $NO_3^-$ concentrations), SAP, green and senesced leaf N and P contents, and N and P resorption efficiency. One-way ANOVA was used to test for variation in the green and senesced leaf N contents and N resorption efficiency of the three dominant plant species, We used stepwise multiple linear regression to examine the relationships between the soil inorganic nitrogen content and green leaf nitrogen content. A similar regression model was established between soil available phosphorus content and green leaf phosphorus content.

Structural equation models (SEMs) were conducted using IBM SPSS Amos version 21 statistical software (Amos Development Co., Armonk, NY, USA). We hypothesize that the pathways responsible for the effects of grazing on N resorption efficiency are influenced by the green and senesced leaf nitrogen contents and soil inorganic nitrogen content. Similarly, we also hypothesize that the effects of grazing on P resorption efficiency is influenced by the green and senesced leaf nitrogen contents and available phosphorus content. A Chi-square ($\chi^2$) test with the associated probability, the root mean square error of approximation (RMSEA) with the associated probability, and the Comparative Fit Index (CFI) were used to evaluate the fitness of the model. Non-significant $\chi^2$ and RMSEA ($P > 0.05$), and CFI > 0.90 indicate the good fit of SEMs. In addition, the significance of each path in the model depends on the probability level ($P < 0.05$).

## RESULTS

### Responses of the key soil resources and plant biomass

Compared with the exclosure, long-term overgrazing significantly decreased the SWC, SOC, STN, and SIN content, whereas the SAP content was significantly enhanced by long-term grazing ($P < 0.01$, Table 1). Some of the key soil resources (SWC, SIN, and SAP) were higher in 2013 than in 2014. The three dominant plant species investigated in this study accounted for >65% of the ANPP in grazed and grazing exclusion plots, respectively (Fig. 1). Long-term grazing significantly decreased the dominant plant species biomass in both years of sampling ($P < 0.05$, Fig. 1). However, *L. chinensis* recorded a higher proportion of biomass in the grazing exclusion plot in both years (41.84% in 2013 and 41.93% in 2014) of sampling, while the biomass of *S. grandis* (36.54%) was higher in the grazed plot in 2014 compared with all other species (Fig. 1E).

**Table 1 Soil characteristics (0–20 cm) across the 30 years grazing and enclosure in a temperate steppe of northern China.**

|  |  | SWC (%) | SOC (g kg$^{-1}$) | STN (g kg$^{-1}$) | STP (g kg$^{-1}$) | SIN (mg kg$^{-1}$) | SAP (mg kg$^{-1}$) |
|---|---|---|---|---|---|---|---|
| Treatment | Enclosure | 9.03$^a$ | 20.49$^a$ | 1.83$^a$ | 0.38 | 22.42$^a$ | 2.96$^b$ |
|  | Grazing | 8.44$^b$ | 17.40$^b$ | 1.44$^b$ | 0.37 | 13.48$^b$ | 4.15$^a$ |
| Year | 2013 | 9.24$^a$ | 19.37 | 1.64 | 0.38 | 19.55$^a$ | 3.86$^a$ |
|  | 2014 | 8.24$^b$ | 18.51 | 1.63 | 0.37 | 16.35$^b$ | 3.24$^b$ |
| *P* value | Y | <0.0001 | 0.071 | 0.735 | 0.53 | <0.0001 | <0.0001 |
|  | T | <0.0001 | <0.0001 | <0.0001 | 0.134 | <0.0001 | <0.0001 |
|  | Y × T | 0.194 | 0.219 | 0.785 | 0.971 | 0.015 | 0.057 |

**Notes.**

SWC, soil water content; SOC, soil organic carbon content; STN, soil total nitrogen content; STP, soil total phosphorus content; SIN, soil inorganic nitrogen; SAP, soil available phosphorus content.

[a,b] Values in the same column with different letters are significantly different ($P < 0.05$).

Y, year; T, treatment.

## Responses of leaf nutrient concentration and N resorption efficiency to long-term grazing

The N (18.48 mg g$^{-1}$) and P (1.25 mg g$^{-1}$) concentrations were higher in the grazed plots compared with the grazing exclusion plots (Table 2). Both the green and senesced leaves N concentrations were higher in 2013 than in 2014 ($P < 0.001$), but there was no difference in N resorption efficiency between the years (Table 2). Nitrogen resorption efficiency was higher in the grazed plot than grazing exclusion plots ($P < 0.001$, Table 2). The P resorption efficiency was higher in 2014 than in 2013 ($P < 0.001$, Table 2).

When we further analyzed the effects of grazing on each species individually, we found that grazing significantly increased the N concentration of *L. chinensis* (21.69 mg g$^{-1}$; $P < 0.001$) and *S. grandis* (18.04 mg g$^{-1}$; $P < 0.001$), but not of *C. squarrosa* (Table 3). Nitrogen resorption efficiency of both *L. chinensis* and *S. grandis* was higher in the grazed plot than grazing exclusion plots ($P < 0.001$, Table 3). In contrast, grazing significantly decreased nitrogen resorption efficiency of *C. squarrosa* ($P = 0.001$, Table 3). Grazing significantly increased the green and senesced leaves P concentrations of the *L. chinensis* ($P < 0.001$, Table 3). Long-term grazing significantly increased the P resorption efficiency of *S. grandis* and decreased that of *C. squarrosa* ($P < 0.001$, Table 3).

Leaf nutritional traits of the three dominant species responded differently to grazing exclusion (Fig. 2). Green leaf N concentration of *L. chinensis* was the highest among the three dominant species ($P < 0.05$, Fig. 2). The green leaf N and P concentrations of *L. chinensis* and *S. grandis* were higher under long-term grazing than under grazing exclusion both in 2013 and 2014 ($P < 0.05$, Figs. 2A, 2B). Long-term grazing significantly decreased senesced leaf N concentration of *L. chinensis* but increased that of *S. grandis* in 2013 and 2014 ($P < 0.05$, Fig. 2C). In contrast, the senesced P leaf concentration of *C. squarrosa* was enhanced by long-term grazing in both years of sampling ($P < 0.05$, Fig. 2D). There was an increase in the N resorption efficiency of *L. chinensis* and *S. grandis* in the long-term grazing plot in 2013 and 2014, whereas a decrease in N resorption efficiency was recorded for *C. squarrosa* in 2014 ($P < 0.05$, Fig. 2E). In 2013, P resorption efficiency was higher in

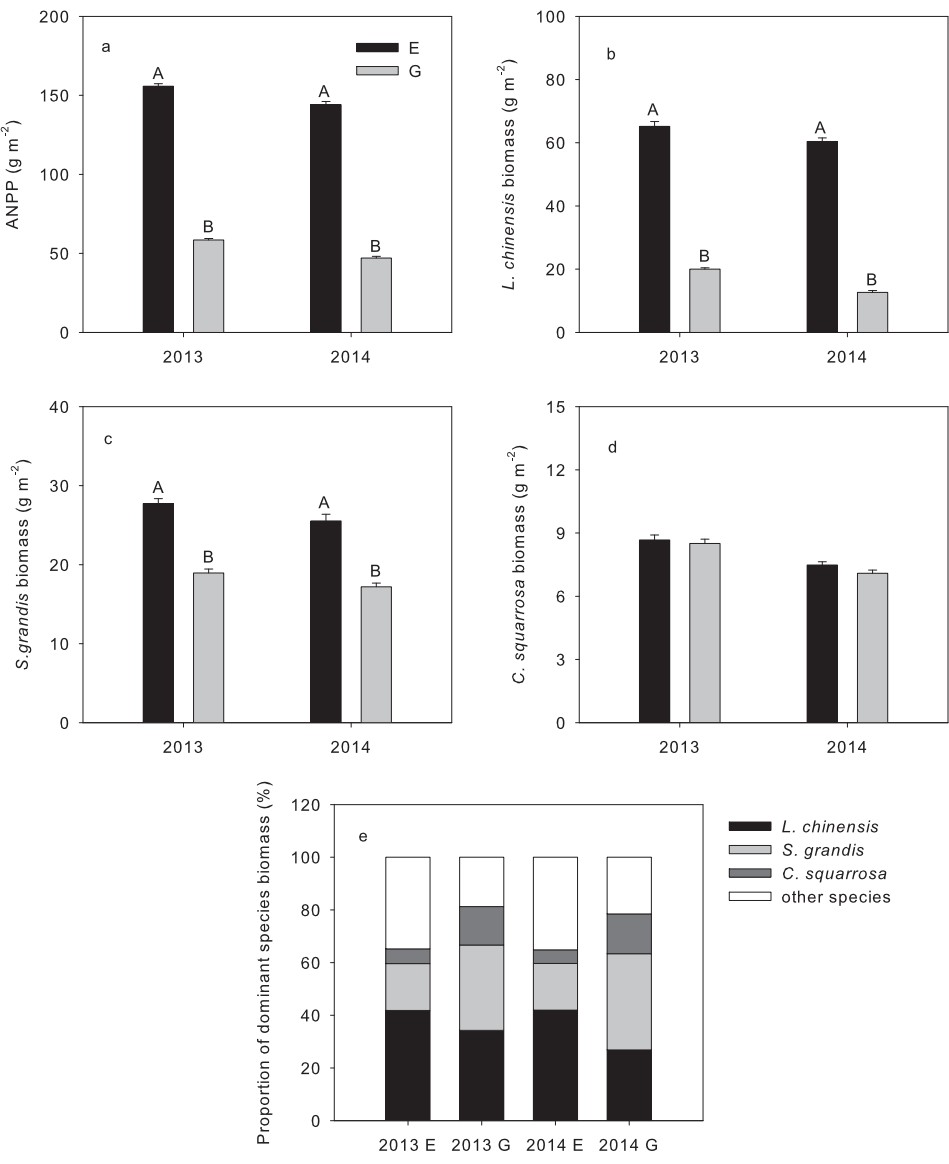

**Figure 1   ANPP, the biomass of three dominant species and proportion of dominant species in the community from typical steppe of Inner Mongolia in grazing exclusion and grazing both in 2013 and 2014.** Above-ground net primary productivity (ANPP) (A), the biomass of *L. chinensis* (B), *S. grandis* (C) and *C. squarrosa* (D) and proportion of dominant species in the community (E) from typical steppe of Inner Mongolia in grazing exclusion (E) and grazing (G) both in 2013 and 2014. Error bars are SE ($N = 15$). Different letters indicate significant differences ($P < 0.05$) between grazing exclusion and grazing for each species.

the grazed plot for *S. grandis* and lower in the same plot for *C. squarrosa* in both years of sampling ($P < 0.05$, Fig. 2F).

**Table 2** Results of three-way ANOVAs on the effects of species (*L. chinensis*, *S. grandis* and *C. squarrosa*), treatment (E: grazing exclusion and G: overgrazing), year (2013 and 2014) and their interactions on N and P concentrations in green leaves (Ng and Pg) and senesced leaves (Ns and Ps), N resorption efficiency (NRE) and P resorption efficiency (PRE) in the typical steppe.

| | | Ng (mg g$^{-1}$) | Pg (mg g$^{-1}$) | Ns (mg g$^{-1}$) | Ps (mg g$^{-1}$) | NRE (%) | PRE (%) |
|---|---|---|---|---|---|---|---|
| Species | *L. chinensis* | 20.69$^a$ | 1.32$^a$ | 7.79$^a$ | 0.60$^b$ | 69.12$^a$ | 64.71$^a$ |
| | *S. grandis* | 17.26$^b$ | 1.11$^c$ | 8.02$^a$ | 0.51$^c$ | 64.23$^b$ | 64.87$^a$ |
| | *C. squarrosa* | 15.75$^c$ | 1.23$^b$ | 6.87$^b$ | 0.73$^a$ | 53.85$^c$ | 48.73$^b$ |
| Treatment | E | 17.33$^b$ | 1.19$^b$ | 7.93a | 0.57$^b$ | 59.62$^b$ | 59.45 |
| | G | 18.48$^a$ | 1.25$^a$ | 7.19$^b$ | 0.65$^a$ | 65.19$^a$ | 59.43 |
| Year | 2013 | 18.43$^a$ | 1.24$^a$ | 7.75$^a$ | 0.64$^a$ | 62.71 | 57.89$^b$ |
| | 2014 | 17.37$^b$ | 1.20$^b$ | 7.37$^b$ | 0.58$^b$ | 62.10 | 60.98$^a$ |
| *P* value | S | <0.0001 | <0.0001 | <0.0001 | <0.0001 | <0.0001 | <0.0001 |
| | T | <0.0001 | <0.0001 | <0.0001 | <0.0001 | <0.0001 | 0.985 |
| | S × G | <0.0001 | <0.0001 | <0.0001 | <0.0001 | <0.001 | <0.0001 |
| | Y | <0.0001 | <0.0001 | <0.0001 | <0.0001 | <0.001 | <0.0001 |
| | S × Y | <0.0001 | 0.049 | <0.0001 | 0.173 | 0.042 | 0.341 |
| | T × Y | 0.722 | 0.165 | 0.041 | <0.0001 | 0.380 | <0.001 |
| | S × T × Y | 0.042 | 0.116 | <0.0001 | 0.944 | 0.528 | 0.101 |

**Notes.**

$^{a,b,c}$Values in the same column with different letters are significantly different ($P < 0.05$).

S, plant species; T, treatment; Y, year.

## Impact of plant and soil properties on N resorption efficiency under long-term grazing

The green leaf N content was positively correlated to SIN content for *L. chinensis* under grazing exclusion ($R^2 = 0.58$, $P < 0.001$; Fig. 3A). In the long-term grazing plot, the green leaf N concentration of *L. chinensis* ($R^2 = 0.24$, $P = 0.007$) and *S. grandis* ($R^2 = 0.24$, $P = 0.002$) were positively correlated to SIN (Fig. 3B) A positive relationship was found between the leaf P contents of *C. squarrosa* and SAP in the grazing exclusion plot ($R^2 = 0.16$, $P = 0.028$, Fig. 3C). Moreover, the green leaf P content of all the three dominant species was positively correlated to SAP (*L. chinensis*: $R^2 = 0.27$, $P = 0.003$; *S. grandis*: $R^2 = 0.42$, $P < 0.001$; *C. squarrosa*: $R^2 = 0.38$, $P < 0.001$) under long-term grazing (Fig. 3D). The SEMs explained 90%, 84%, 61%, 83%, 84% and 90% of the variance in N and P resorption efficiency of *L. chinensis*, *S. grandis* and *C. squarrosa*, respectively (Figs. 4A, 4F). Our results showed that grazing has a direct effect on N resorption efficiency of *L. chinensis* and *C. squarrosa*, respectively (Figs. 4A, 4C). The direct and indirect effects of grazing on green leaf nitrogen contents (Ng) and senesced leaf nitrogen contents (Ns) altered the N resorption efficiency of *S. grandis* (Fig. 4B), but P resorption efficiency of the plant was influenced by the direct of grazing on green leaf phosphorus contents (Pg) only *S. grandis* (Fig. 4E). The P resorption efficiency of *L. chinensis* was changed by the indirect effects of grazing on Pg and Ps (Fig. 4D). Similarly, the change in P resorption efficiency of *C. squarrosa* was driven by the indirect effects of grazing on Pg and Ps (Fig. 4F).

**Table 3** Results of two-way ANOVAs on the effects of treatment (E: grazing exclusion; G: overgrazing), year and their interaction on nitrogen and phosphorus concentration in green leaves (Ng and Pg) and senesced leaves (Ns and Ps), nitrogen resorption efficiency (NRE) and phosphorus resorption efficiency (PRE) for each species in the typical steppe.

| | | Treatment | | Year | | P value | | |
|---|---|---|---|---|---|---|---|---|
| | | E | G | 2013 | 2014 | T | Y | T × Y |
| *L. chinensis* | Ng | 19.69[b] | 21.69[a] | 21.57[a] | 19.81[b] | <0.0001 | <0.0001 | 0.720 |
| | Ns | 8.91[a] | 6.66[b] | 7.78 | 7.80 | <0.0001 | 0.808 | <0.0001 |
| | Pg | 1.26[b] | 1.39[a] | 1.34[a] | 1.30[b] | <0.0001 | <0.0001 | 0.502 |
| | Ps | 0.55[b] | 0.63[a] | 0.62[a] | 0.58[b] | <0.0001 | 0.022 | 0.021 |
| | NRE | 63.26[b] | 74.99a | 69.21 | 69.04 | <0.0001 | 0.713 | 0.005 |
| | PRE | 63.96 | 65.47 | 63.70 | 65.73 | 0.828 | 0.157 | 0.287 |
| *S. grandis* | Ng | 16.48[b] | 18.04[a] | 17.90[a] | 16.62[b] | <0.0001 | <0.0001 | 0.084 |
| | Ns | 7.87 | 8.18 | 8.11 | 7.94 | 0.085 | 0.360 | 0.467 |
| | Pg | 1.09[b] | 1.14a | 1.12 | 1.10 | <0.0001 | 0.064 | 0.029 |
| | Ps | 0.50 | 0.52 | 0.54a | 0.48[b] | 0.102 | <0.0001 | 0.003 |
| | NRE | 60.15[b] | 68.32[a] | 64.64 | 63.84 | <0.0001 | 0.416 | 0.764 |
| | PRE | 61.55[b] | 68.20a | 62.66[b] | 67.09a | <0.0001 | <0.0001 | <0.0001 |
| *C. squarrosa* | Ng | 15.81 | 15.69 | 15.82 | 15.68 | 0.356 | 0.257 | 0.148 |
| | Ns | 6.72[b] | 7.02[a] | 7.37[a] | 6.37[b] | 0.046 | <0.0001 | 0.448 |
| | Pg | 1.22 | 1.24 | 1.25[a] | 1.21[b] | 0.074 | <0.0001 | 0.387 |
| | Ps | 0.66[b] | 0.79[a] | 0.77[a] | 0.69[b] | <0.0001 | <0.0001 | 0.015 |
| | NRE | 55.46[a] | 52.25[b] | 54.28 | 53.32 | 0.001 | 0.358 | 0.221 |
| | PRE | 52.83[a] | 44.63[b] | 47.32[b] | 50.13[a] | <0.0001 | 0.021 | 0.016 |

**Notes.**

[a,b]Values in the same column with different letters are significantly different (*P* < 0.05).

T, treatment; Y, year.

# DISCUSSION

## Effects of overgrazing on the leaf nutrient status

The findings from this study reveal that overgrazing significantly increased the N concentration in the green leaves of the dominant species investigated. This corroborates earlier reports (*Bai et al., 2012*; *Ma et al., 2019*) that herbivore grazing modifies plant N concentration. Compared with the grazing exclusion plot, the high green leaf N concentration recorded for the dominant species (*L. chinensis, S. grandis,* and *C. squarrosa*) in the long-term grazing plot may be attributed to increased synthesis of protein by the ribosome to aid plant growth after successive grazing (*Elser et al., 2000*; *Ma et al., 2019*). This suggests a positive response of the species under study to grazing activities by livestock. In addition, the increased green leaf N concentrations under long-term grazing potentially improve the plant species compensatory growth to reduce biomass loss as found by *Wang et al. (2004)*. Moreover, the positive correlation between green leaf N concentration and soil inorganic N content in the grazed plot indicates that the return of livestock excreta (i.e., faeces and urine) could have stimulated root exudation of C-rich substances required for increased N mineralization to enhance green leaf N concentration (*Bai et al., 2012*). Therefore, our result supports the idea that an increase in leaf N concentration of plants

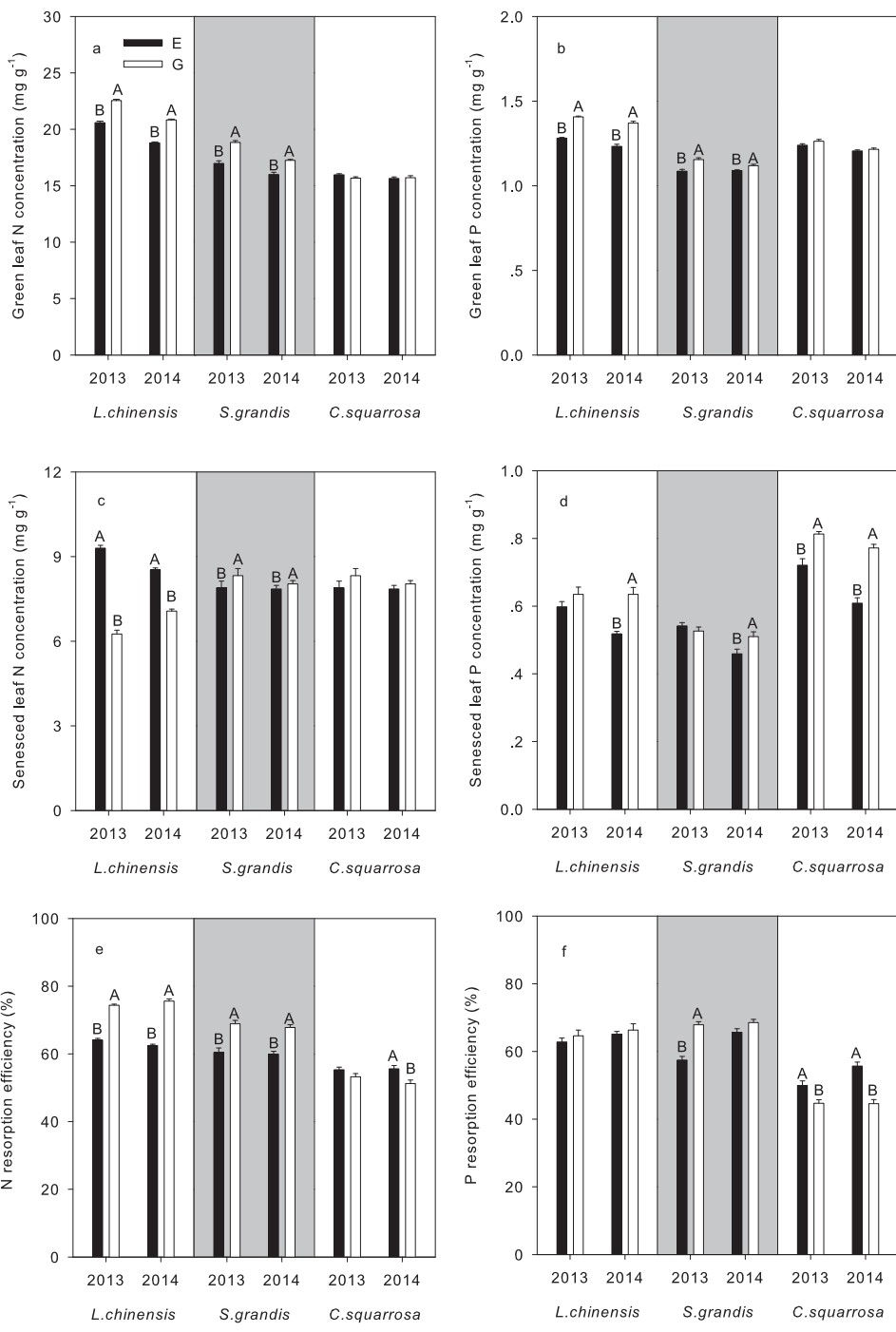

**Figure 2  Nitrogen concentrations, nitrogen and phosphorus resorption efficiency of three dominant species from typical steppe of Inner Mongolia in grazing exclusion and overgrazing both in 2013 and 2014.** Nitrogen concentrations in green (A, B) and senesced (C, D) leaves and nitrogen and phosphorus resorption efficiency (E, F) of three dominant species from typical steppe of Inner Mongolia in grazing exclusion (E) and overgrazing (G) both in 2013 and 2014. Error bars are SE ($N = 15$). Different letters indicate significant differences ($P < 0.05$) among treatments for each species.

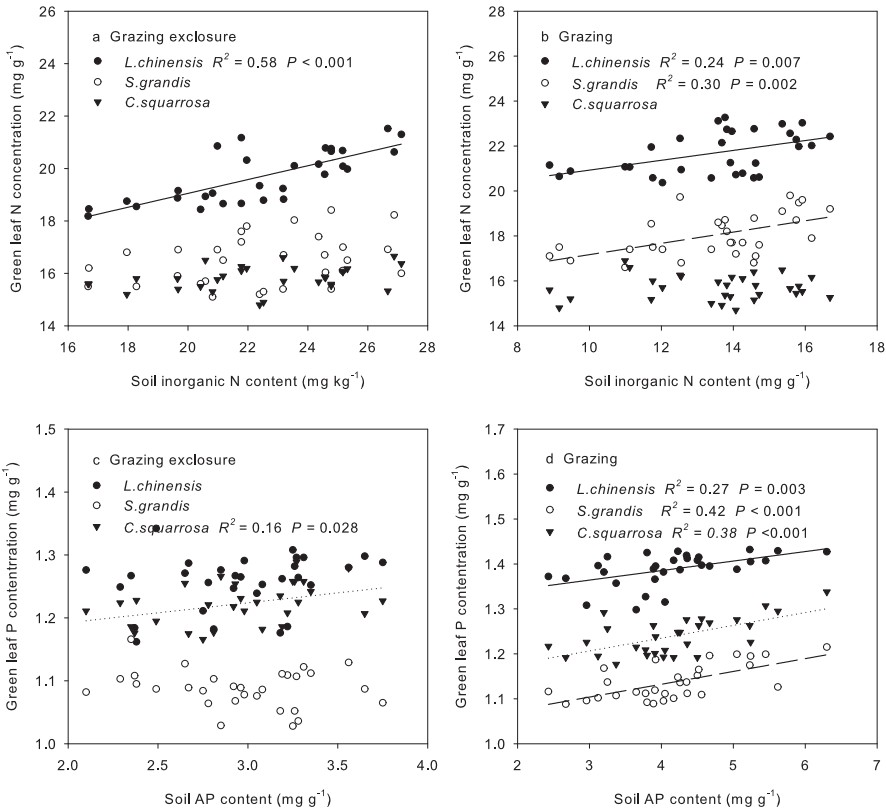

**Figure 3** Spatial dependence of green leaf nitrogen concentrations on soil inorganic N content, and spatial dependence of green leaf phosphorus content on soil available phosphorus content. Spatial dependence of green leaf nitrogen concentrations on soil inorganic N content in exclusion (A) and grazing (B), and spatial dependence of green leaf phosphorus content on soil available phosphorus content in grazing exclusion (C) and overgrazing (D) across the 30 plots in two years.

is an important mechanism of responding to grazing in grassland ecosystems (*Elser et al., 2010*; *Peñuelas et al., 2013*). In the grazing exclusion plot, only *C. squarrosa* recorded a weak positive relationship between its green leaf P content and SAP. Conversely, the green leaf P content of all the dominant species are positively related to SAP in the long-term grazing plot. Interestingly, SAP is higher in the long-term grazing than the grazing exclusion plot. This implies that the decomposition of faeces and urine in the long-term grazing plot enhanced the faster release of P into the soil (*Ma et al., 2019*), which makes more P available through microbial mineralization for plants uptake (*Chen et al., 2004*). Therefore, grazing can increase green leaf P concentration indirectly.

## Effects of overgrazing on leaf nutrient resorption

Although the return of plant foliage into the soil as organic matter potentially increases the N and P input into the ecosystem, the low nutrient resorption efficiency of *C. squarrosa* in the long-term grazing plot may be related to a faster rate of nutrient cycling to replenish plant resources lost to grazing. The reduction in SWC, SOC, and STN recorded in our study

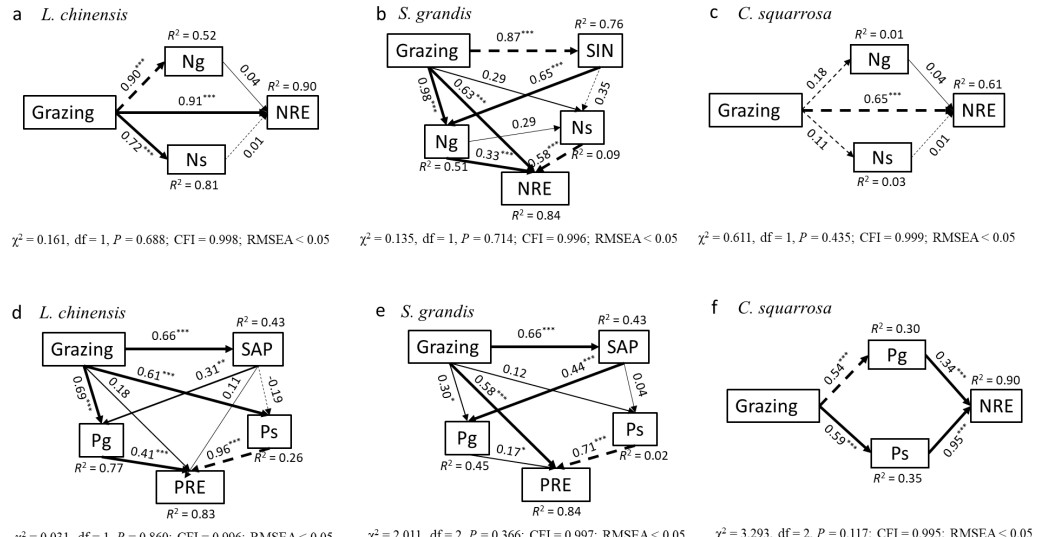

**Figure 4** **Path analyses on the impacts of and path analyses on the impacts of overgrazing on nitrogen resorption efficiency and phosphorus resorption efficiency of three dominant plant species.** Path analyses on the impacts of and path analyses on the impacts of overgrazing on nitrogen resorption efficiency (A, B, C) and phosphorus resorption efficiency (D, E, F) of three dominant plant species (*L. chinensis*; *G. grandis*; *S. squarrosa*). Solid and dashed arrows represent significant ($P < 0.05$, marked *; $P < 0.01$, marked **; $P < 0.001$, marked *** in the figure) and non-significant ($P > 0.05$) paths. Values associated with arrows represent standardized path coefficients. SIN, soil inorganic nitrogen content; Ng, nitrogen concentrations of green leaves; Ns, nitrogen concentrations of senesced leaves; NRE, nitrogen resorption efficiency; SAP, available phosphorus content; Pg, phosphorus concentrations of green leaves; Ps, phosphorus concentrations of senesced leaves.

is consistent with the report by *Millett & Edmondson (2015)*, and this further strengthens the observed grazing-induced changes in nutrient resorption efficiency of the plant species. *Millett et al. (2005)* showed that low levels of simulated herbivore grazing had no impact on the N resorption efficiency of *Betula pubescens*, but *Silla & Escudero (2003)* found that herbivory (light grazing) reduced the green leaf N concentration of Mediterranean *Quercus* species with consequent effect on leaf N resorption efficiency. In this study, we found that more than 50% of N is reabsorbed during leaf senescence, suggesting that nutrient resorption is one of the key mechanisms used by plants for nutrient conservation under grazing. Our result indicates that leaf N resorption efficiency is influenced by grazing and different among the dominant species.

With respect to plant nutrient resorption efficiency, our results suggest that the divergent response of leaf N and P resorption efficiency pattern to overgrazing between the three species most likely resulted from the differences in biological features and resource-use strategies. Our results indicate that *L. chinensis*, *S. grandis* and *C. squarrosa* exhibit a different pattern of N and P resorption efficiency under long-term grazing and grazing exclusion. We surmise that species with high leaf N and P concentrations generally have a relatively fast metabolism because of their high growth rates (*Elser et al., 2003*). Compared with species with low nutrient concentrations, these species with high leaf N

and P concentrations need to obtain more resources (*Tilman, 1982*). Our findings also suggest that *L. chinensis* relies more on plant internal nutrient cycling in overgrazing and more on soil nutrient resources that are still stored in the grassland. Therefore, *L. chinensis* shows high N resorption efficiency for under overgrazing. Compared with *L. chinensis* and *S. grandis*, it takes an additional 6 weeks for *C. squarrosa* to develop after grazing, particularly when the temperatures are more favorable for their growth (*Liang, Michalk & Millar, 2002*). *C. squarrosa* is more resistant to grazing than *L. chinensis* and *S. grandis* based on avoidance and tolerance traits (*Zheng et al., 2011*). It is noteworthy, however, that compared with $C_4$ grasses (e.g., *C. squarrosa*), $C_3$ grasses (e.g., *L. chinensis* and *S. grandis*) generally maintain green leaves longer in regions with cold winters and may stay green in cold regions (*Chamaillé-Jammes & Bond, 2010*; *Fanselow et al., 2011*). More importantly, the early onset of leaf senescence for *C. squarrosa* in the region of study (i.e., short plant growing period from early June to the end of September) results in nutrient sequestration in dead leaves. The dead leaves of *C. squarrosa* are easily dispersed by wind, leading to a potential nutrient loss in the soil. As a result, grazing decreased leaf N and P resorption efficiency.

In addition, the preference of livestock for different forage species within the grazing environment is related to the observed changes in the nutrient resorption efficiency in the long-term grazing treatment. *L. chinensis* is highly relished by livestock when at the same stage of growth as other grassland species, due to its higher nitrogen concentration and low C/N ratio. Continuous grazing across the seasons of the year decreased the biomass of *L. chinensis* in both years of sampling. Further, soil cannot provide sufficient nutrients to plants requiring a large amount of N for growth. This may have contributed to the increased N resorption efficiency recorded for *L. chinensis* under long-term grazing. This finding agrees with the results of *Killingbeck (1996)* that plants with high nutrient resorption efficiency tend to have a lower dependence on the soil nutrient pool. Our SEMs also show that overgrazing has a direct effect on the leaf N resorption efficiency of *L. chinensis*. *S. grandis* is usually abundant in drier and N-limited conditions (*Chen et al., 2005*), while herbivores prefer *L. chinensis* to the former grass. This implies that under long-term grazing condition, *S. grandis* has a better advantage to efficiently utilize nutrients in a nutrient-limited soil, and SIN content plays a vital role in driving the green leaf N concentration of *S. grandis*. The SEMs also showed that green and senesced leaf N concentration play an important role in driving N resorption efficiency under long-term grazing.

Our results show that grazing had strong direct effects on P resorption efficiency of *S. grandis*, but not on P resorption efficiency of *L. chinensis*. *L. chinensis* has a high degree of stoichiometric N:P homeostasis (grazing 15.60 vs grazing exclusion 15.63) compared with *S. grandis* (grazing 15.12 vs grazing exclusion 15.82) in the green leaves (Table 3), which may in part be supported by previous studies in semiarid steppe (*Yu et al., 2010*; *Mariotte et al., 2017*). In addition, phosphorus may not be a limiting factor under overgrazing. The increase in soil P concentration could actually result in increased P availability. The consistent change in leaf P concentration of *L. chinensis* in the green leaf and the senesced leaf may result in no change in phosphorus recovery efficiency. The enhanced P resorption efficiency by *S. grandis* accounts for the increased plant-available P in the soils under

long-term grazing. This is corroborated by the positive linear relationship between SAP content and the green leaf P concentration of *S. grandis*. SEM also showed that the indirect effect of grazing (via changes in green leaf P concentration of *S. grandis*) is a major driver that changes P resorption efficiency of *S. grandis*. The high green leaf P concentration and the inconsistent change in senesced leaf P concentration of *S. grandis* under long-term grazing may have resulted in its increased P resorption efficiency. Our SEM result suggests that senesced leaf P concentration is the predominant factor that determines P resorption efficiency of *C. squarrosa*. Previous studies (e.g., *Osborne, 2008*) have shown that $C_3$ and $C_4$ plant species differ in their sensitivity to frost, where $C_4$ plant species had higher green leaf mortality than $C_3$ plants after a frost event. In this study, leaf senescence sets in earlier in $C_4$ plant species (*C. squarrosa*, late September) than in $C_3$ plant species (*L. chinensis* and *S. grandis*, mid-October), which leads to the incomplete P resorption of *C. squarrosa*. Consequently, the nutrient is retained in the plant leaves, and as such, long-term grazing decreased the leaf P resorption efficiency of *C. squarrosa*.

## CONCLUSIONS

This study evaluates how long-term overgrazing affects plants' leaf nutrient status and resorption in three dominant species of northern China. We found that overgrazing increases green leaf nutrient concentrations and enhances ecosystem nutrient cycling in the ecosystem through increasing senesced leaf nutrient concentrations. Overgrazing increased leaf N concentration and N resorption in *L. chinensis* and *S. grandis*, but overgrazing only had strong effects on P resorption efficiency in *S. grandis*. In contrast, overgrazing reduced N and P resorption efficiency of *C. squarrosa*. Our results provide a better understanding of plant internal nutrient retranslocation in response to grassland management. Our studies suggest that the responses of the three dominant plant species nutrient resorption efficiency to overgrazing appear to be species dependent and associated with species differences in physiological characteristics and adaptive strategies.

## ACKNOWLEDGEMENTS

Authors would like to thank Inner Mongolia University and the Inner Mongolia Grassland Ecosystem Research Station of the Chinese Academy of Science for providing the field permit for sample processing. We thank for Shiming Tang, School of Ecology and Environment at Inner Mongolia University, for helping set up the experiment and collect plant samples.

### Funding

This study was financially supported by the National Natural Science Foundation of China (31770542, 41601269), the Major Project of National Natural Science Foundation of Inner Mongolia (2020ZD06), Ningxia Hui Autonomous Region key research and development plan project (2017BY085), the National Natural Science Foundation of Inner Mongolia

(2019MS03001, 2019MS03003), the National Natural Science Foundation of China and USA (31761123001-1), the Central Nonprofit Research Institutes Fundamental Research Funds (1610332020005), the Inner Mongolia Science and Technology Plan (Grant No. 2019GG009) and Grass and Livestock Resource-Saving Production System and Sustainable Development Mode of Ecologically Vulnerable Areas ([2018]1351). The funders had no role in study design, data collection and analysis, decision to publish, or preparation of the manuscript.

### Grant Disclosures

The following grant information was disclosed by the authors:
The National Natural Science Foundation of China: 31770542, 41601269.
The Major Project of National Natural Science Foundation of Inner Mongolia: 2020ZD06.
Ningxia Hui Autonomous Region key research and development plan project: 2017BY085.
The National Natural Science Foundation of Inner Mongolia: 2019MS03001, 2019MS03003.
The National Natural Science Foundation of China and USA: 31761123001-1.
The Central Nonprofit Research Institutes Fundamental Research Funds: 1610332020005.
The Inner Mongolia Science and Technology Plan: 2019GG009.
Grass and Livestock Resource-Saving Production System and Sustainable Development Mode of Ecologically Vulnerable Areas: [2018]1351.

### Competing Interests

Saheed Olaide Jimoh is a volunteer researcher at the Sustainable Environment Food and Agriculture Initiative (SEFAAI). The authors declare there are no competing interests.

### Author Contributions

- Zhen Wang conceived and designed the experiments, performed the experiments, analyzed the data, prepared figures and/or tables, authored or reviewed drafts of the paper, and approved the final draft.
- Saheed Olaide Jimoh performed the experiments, analyzed the data, authored or reviewed drafts of the paper, and approved the final draft.
- Xiliang Li analyzed the data, authored or reviewed drafts of the paper, and approved the final draft.
- Baoming Ji and Paul C. Struik analyzed the data, prepared figures and/or tables, authored or reviewed drafts of the paper, and approved the final draft.
- Shixian Sun conceived and designed the experiments, performed the experiments, prepared figures and/or tables, authored or reviewed drafts of the paper, and approved the final draft.
- Ji Lei performed the experiments, authored or reviewed drafts of the paper, and approved the final draft.
- Yong Ding performed the experiments, prepared figures and/or tables, and approved the final draft.
- Yong Zhang performed the experiments, prepared figures and/or tables, authored or reviewed drafts of the paper, and approved the final draft.

## Field Study Permissions

The following information was supplied relating to field study approvals (i.e., approving body and any reference numbers):

The study area was conducted at the Inner Mongolia Grassland Ecosystem Research Station (43°38′N, 116°42′E)

## Data Availability

The raw measurements available in the Supplemental Files.

## Supplemental Information

Supplemental information for this article can be found online at http://dx.doi.org/10.7717/peerj.9915#supplemental-information.

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
