# Peer review of "Different responses of plant N and P resorption to overgrazing in three dominant species in a typical steppe of Inner Mongolia, China"

_PeerJ, doi:10.7717/peerj.9915_

## Round 0.1 · original submission · Major Revisions

Dear authors

Two reviews have been received which agreed in that the manuscript is interesting however it needs major revisions to be improved.

Please do send a new version with all changes suggested by reviewers taken into account.

Best regards

Reviewer 1 ·

Basic reporting

The language is a bit redundant, not standard English, and at quite many occasions, there are repetitions especially in Results section, e.g., the sentence in Line 229-230 is repetitive to the sentence in 208-209, and also to line 218-219. The article is self-contained, and has included sufficient introduction and background to demonstrate how the work fits into the broader field of knowledge. The structure of the manuscript conform to the acceptable format of "standard sections"

Experimental design

The manuscript presented an Original primary research within Aims and Scope of the journal. Research questions were well defined,relevant and meaningful, but with improper expression therein: in the first question, "changes in" should be deleted, because what the overgrazing altered is N and P concentration and resorption efficiency, as presented in the results section, not their changes. Moreover, it was stated (Line 54-55) how the study fills an knowledge gap.
Methods described with sufficient detail & information to replicate. However, the investigation fall short of the pseudo-replication in the experimental plots designing, which is the biggest problem of the study.

Validity of the findings

Some results are contradictory. e.g., Line 218-220, here it was mentioned that grazing decreased NRE of C. squarrosa, but in line 212, it was meant that grazing increased its NRE.
The conclusion is merely repetition of results.

Additional comments

The topic of overgrazing effect on nutrient resorption is interesting and of significance for nutrient cycling research and grassland management. However, My major concerns include the pseudo-replication of the experimental designing. Another one is that, In the discussion and Conclusion, the authors claimed that leaf N and P in the green and senesced leaves determines the nutrient resorption efficiency, but that was not the underlying mechanisms in terms of biological or ecological processes, but such determination was already in the formula calculating the nutrient resorption efficiency, i.e, it is just mathematical determination. I suggest the authors to discuss the discrepancies between the three species in nutrient resorption from the perspectives of their difference in biological features and adaptive strategies, thereby validate the inclusion of these three plant species in this article. By the way, in the Method section, you should explain why you choose these three species, and introduce a bit the biological features of the three species as a background information.

Reviewer 2 ·

Basic reporting

no comment

Experimental design

no comment

Validity of the findings

no comment

Additional comments

General comments:
Wang et al. examined the resorption of N and P by plants of three grassland species in two contrasting communities (overgrazing site vs. grazing exclosure) in Inner Mongolia grassland. They found that grazing resulted in increases in the concentrations of N and P in green leaves and in the leaf N resorption efficiencies of Leymus chinensis and Stipa grandis but not in Cleistogenes squarrosa. They demonstrated that grazing was the primary factor leading to changes in N resorption efficiency for three species but only directly affected P resorption efficiency for S. grandis. They concluded that effects of grazing on different species were associated with their dynamics nutrient resorption efficiency due to variation in forage quality.
Overall, I think this is an interesting study on an important issue about how grazing affects the nutrients cycling processes in grassland ecosystems. The manuscript was well organized. Data analysis seems to be adequate, and results are well presented. However, the Discussion section needs to be improved.
I have several concerns on the Discussion and Conclusion Sections. First, the leaf concentrations and resorption efficiencies of N in Leymus chinensis and Stipa grandis were enhanced by grazing. This is an important finding of this study because these two species are widely distributed in Eurasia steppe, the largest grassland in the world, however, the explanation for this point is not sufficient, especially not associated with the physiological characteristics of these species. Second, based on the results of structure equation model analysis, grazing had strong direct effects on N and P resorption efficiency of S. grandis, but only had strong direct effects on N not on P resorption efficiency of L. chinensis. Why? They should give some explanation, at least some possible mechanisms, for these differences in Discussion Section. Third, in the end of Abstract, they suggested that effects of grazing on different species are associated with their dynamics nutrient resorption efficiency due to variation in forage quality. This conclusion has not sufficient evidence support by the results in this study. They did not examine the forage quality of each species. For the conclusion sentence, they should focused on the new findings of this study rather than over explanation. Finally, they used too many abbreviations that restricts the understanding by a reader.

Minor points:
Line 26: enclosure or exclosure ? please use the same word.
Line 29: C. squarossa, here you should use the full Latin name.
Line 86: three plant groups or three plant species?
Line 122: how did you place four 1 x 1 m2 quadrats in three 1.5 x 1.5 m cages?

---

## Round 0.2 · accepted · Accept

Dear Authors,

Based on my reading of the new version of the manuscript and your rebuttal letter, I have decided to accept this nice contribution to Peer J. If you continue this type of work please do not mix individuals in your samples (the reallocation could be very different between individuals within species) in my opinion this is the most important problem of the study but still is worth to be published. I understand that the pseudoreplication issue is just inherent to the type of question and setting, on the other hand, taking advantage of such a long term exclosure is great! and I loved the topic!

Congratulations